# New Sagittal and Vertical Cephalometric Analysis Methods: A Systematic Review

**DOI:** 10.3390/diagnostics12071723

**Published:** 2022-07-15

**Authors:** Jacek Kotuła, Anna Ewa Kuc, Joanna Lis, Beata Kawala, Michał Sarul

**Affiliations:** 1Department of Dentofacial Orthopedics and Orthodontics, Wroclaw Medical University, Krakowska 26, 50-425 Wroclaw, Poland; joanna.lis@umed.wroc.pl (J.L.); beata.kawala@umed.wroc.pl (B.K.); michal.sarul@umed.wroc.pl (M.S.); 2Dental Star Specjalistyczne Centrum Stomatologii Estetycznej, 15-215 Białystok, Poland; dental.star@wp.pl

**Keywords:** cephalometric analysis, sagittal discrepancy, vertical discrepancy

## Abstract

Cephalometric analysis is an essential tool used in orthodontic diagnosis and treatment planning. The main objectives of correct cephalometric analysis include resolving anteroposterior and vertical maxillary and mandibular base discrepancies. For a diagnostic tool to be of value, it should be precise, reliable and reproducible. Unfortunately, according to some studies, the accuracy of input and, therefore, the diagnostic reliability of some of the points and measurements may not be satisfactory. To this end, new cephalometric measurements are being developed with increased precision. In order to properly and definitively determine the usefulness of a given measurement in cephalometric diagnosis, it is necessary to carry out a critical evaluation of available studies. The aim of this systematic review was to evaluate the available scientific literature describing new landmarks and reference linear and angular measurements of 2D cephalometric analyses assessing the sagittal and vertical discrepancy in the position of jaw bases since the last systematic review in 2013. The secondary aim was to assess the accuracy and reliability of new anthropometric landmarks and reference planes in relation to those used previously, and their instability in relation to growth and orthodontic tooth movements. To carry out the intended plan, electronic databases such as PubMed, Scholar Google, Web of Science and Pro Quest were searched using specific keywords. Initially, a total of 1451 articles were retrieved. Then, duplicate articles in all databases were excluded from the resulting publications. The results showed that despite such a high number of articles published in peer-reviewed scientific journals, only 12 studies on new cephalometric analyses in the sagittal plane and 4 studies on new cephalometric analyses in the horizontal plane met the criteria and, as a result, were included in the review.

## 1. Introduction

Methods of conducting the cephalometric analysis of lateral head radiographs in two-dimensional visualisation have been developed since their pioneering use by Broadbent in the USA and Hofrath in Germany that took place in 1931 [1,2,3,4,5,6,7,8]. Since then, cephalometric analysis has been one of the basic instruments used routinely in the diagnosis and planning of orthodontic treatment [3]. Although only a small percentage of orthodontic treatment plans are modified [9] on the basis of cephalometric analysis [1], its results allow the orthodontist to plan a comprehensive therapeutic process that is improved [2]. A correct cephalometric analysis is particularly important in borderline cases when an extraction or orthognathic/surgical treatment plan is considered [4].

Admittedly, modern orthodontics is increasingly using CBCT imaging. However, given the number of scientific studies on 2D cephalometric analyses and the need for the radiological protection of the patient, 2D lateral cephalograms remain the primary diagnostic examination in orthodontic assessment and treatment planning.

To date, many new analyses have been developed, each containing some new measurements and/or reference values [2]. Despite the numerous papers published in peer-reviewed scientific journals in this field [1], the actual value of this imaging technique in orthodontic treatment planning has not been scientifically proven. This is mainly due to the instability of the reference points used in cephalometric diagnosis in relation to changes in growth type and the therapeutic process used. Due to the inaccuracy and differences in the interpretation of the positions of numerous landmarks in 2D lateral cephalograms, new landmarks are being sought that will not change location during the growth process or as a result of tooth movement during treatment. Examples of such measurements include:YEN angle formed by the points S, M, G defining the sagittal relationship between the maxilla and the mandible, first described by Neel et al. in 2009 [10].Pi analysis referring to the angular measurement Pi (GG`M) and the linear measurement Pi (G`M`) based on the points G, M from which the perpendicular goes to the true horizontal plane in the natural position of the head, defining the sagittal relationship between the maxilla and the mandible [11], first described by Kumar et al. in 2012 [12]W angle formed by the points S, M, G, defining the sagittal relationship between the maxilla and mandible, first described by Bhad et al. in 2011 [13]SAR angle formed by the points M, G, W, defining the sagittal jaw base discrepancy, described by Sonahita et al. in 2015 [14]DW angle using Walker and Wing (WW) points to assess the sagittal discrepancy, described by Hatewar et al. in 2015 [15]Tau angle formed by the points T, M, G, defining the sagittal relationship between the maxilla and the mandible, first described by Gupta et al. in 2020 [6]R angle formed by the points N, C, Me to assess the vertical discrepancy, first described by Rizwan and Mascarenhas in 2013 [16]KP (extraoral) plane and points NS, SAE bilaterally to assess the vertical discrepancy, first described by Kattan et al. in 2018 [17]Superior border of the zygomatic arch to assess the vertical discrepancy as an alternative to the Frankfurt horizontal line introduced by Park et al. in 2019 [18]

However, to confirm the diagnostic effectiveness of the above-mentioned measurements, it is necessary to carry out a thorough analysis and comparison of studies on their use. The aim of this study was to evaluate the available scientific literature describing new landmarks and reference linear and angular measurements of 2D cephalometric analyses assessing the sagittal and vertical discrepancies in the positions of jaw bases.

## 2. Methods Protocol and Registration

The protocol for this review was registered on the International Prospective Register of Systematic Reviews (PROSPERO) database (CRD…) available from https://www.crd.york.ac.uk/prospero … (accessed on 1 February 2022). The present systematic review was conducted according to the Preferred Reporting Items for Systematic Reviews and Meta-Analyses (PRISMA) [19] and the Cochrane Handbook for Systematic Reviews of Interventions [20]. The PRISMA flow diagrams summarize all steps in the selection of included studies, were built using an online tool [21] and included eligibility criteria and study participant characteristics. The eligibility criteria for the included studies were defined considering the PICO strategy. The types of studies included in the systematic review were randomized controlled trials, nonrandomized clinical trials, and observational studies. Case reports, case series, letters, comments, short communications, pilot studies (ten patients or fewer), animal studies, in vitro studies, in silico studies, and literature reviews were excluded. The eligible studies were full-text articles in English and Russian with publication date during 2009 to 2021.

## 3. Information Sources and Search Strategy

The search was performed in the electronic databases PubMed and Web of Science; Scholar Google and Pro Quest were used for the identification of the registers and protocols for the clinical trials. The manual search was achieved through examining the bibliographical references of the studies included in the review. This search was carried out from September 2021 to December 2021. The keywords and algorithms used for the search strategy are shown in Table 1. Two reviewers (JK and PS) performed the search and selection. In the absence of unanimity between the researchers, MS had the final say.

## 4. Materials and Methods

To prepare the systematic review, the electronic databases PubMed, Google Scholar, Pro Quest, and Web of Science were searched to find publications that met the inclusion criteria. No attempt was made to explore informally published articles, conference materials or abstracts of presentations given at scientific conferences. The search was conducted from 2013, the end of the previous systematic review, to 2021, extended backward to 2009 to include analyses published between 2009 and 2013 that were not included in the previous publication.

## 5. Selection of Material

Articles were included in this publication in two stages. In the first stage, two orthodontist reviewers (JK and PS) independently reviewed PubMed, Scholar Google Pro Quest, and Web of Sciences using keywords corresponding to the criteria specified in the paper (Table 1). Studies eligible for inclusion in the systematic review were determined by the title and abstract of each record identified by the search.

Then, in the retrieved database of articles, two reviewers made an initial selection of articles that met the search criteria in accordance with the research topic. The following were used as exclusion criteria:Publications in languages other than English and Russian;Publications published before 2009;Publications that appeared repeatedly in various databases;Publications whose full texts were not made available online;Publications evaluating soft tissue analysis.

After initial screening, abstracts of the retrieved publications were analysed and categorised by research topic by each of the two reviewers. At this stage, publications were excluded according to the following criteria:Article objectives were irrelevant to the subject of this review;Articles covered the topic of cone–beam computed tomography;Articles were related to three-dimensional analysis.

Each article included in the next selection stage had to be favourably evaluated by at least one of the reviewers; in the absence of unanimity, the third reviewer had the casting vote. At this stage, full texts of articles were downloaded and subjected to critical analysis. Bibliographies and reference lists of publications that were considered relevant in the first stage were searched manually. The aim of this review was to evaluate parent studies. A detailed selection tree is shown in Figure 1a,b [11].

The collected articles were subjected to risk of bias analysis according to Liu et al. [22] (Table 2a relative to Q1 and Table 2b relative to Q2).

The quality and internal relevance (level of reliability) of each publication were rated as high, moderate or low according to the criteria indicated in a review by Durão et al. [1].

**Table 2 diagnostics-12-01723-t002:** (**a**) The risk-of-bias analysis of articles evaluating new cephalometric analysis parameters in relation to the sagittal plane. (**b**) The risk-of-bias analysis of articles evaluating new cephalometric analysis parameters in relation to horizontal plane.

**(a)**
**Q1**	**Author (year)**	**Neela 2009** [10]	**Bhad 2011** [13]	**Kumar 2012** [12]	**Kumar 2014** [23]	**Sonahita, A.; 2014** [14]	**Hatewar 2015** [15]	**Ali, S.M.; 2018** [24]	**Ahmed 2018** [25]	**Shetty****2019** [26]	**Gupta 2020** [6]	**Jedliński 2020** [7]	**Gokhan 2021** [8]
A confounding												
B selection bias												
C classification of interventions												
D deviations from intervention												
E missing data												
F measuring the results												
G reporting bias												
H overall												
**(b)**
**Q2**	**Author (year)**	**Rizwan 2013** [16]	**Ahmed M. 2016** [25]	**Kattan EE.****2018** [17]	**Park JA. 2019** [18]
A confounding				
B selection bias				
C classification of interventions				
D deviations from intervention				
E missing data				
F measuring the results				
G reporting bias				
H overall				

O Yellow unclear risk. O Green—low risk.

### 5.1. Levels of Evidence and Criteria for Synthesising Evidence

#### 5.1.1. High Level of Evidence

Research was classified as having a high level of evidence if it met all of the following criteria:An independent blind comparison between test and reference methods was performed (in Table 3a relative to Q1 and Table 3b relative to Q2, marked as A).Population was described in such a way that the condition, prevalence and severity of the condition were clear. The spectrum of patients was similar to the spectrum of patients on whom the research method would be used in clinical practice (in Table 3a relative to Q1 and Table 3b relative to Q2, marked as B).The test method results did not influence the decision to perform reference method (in Table 3a relative to Q1 and Table 3b relative to Q2, marked as C).The test and reference methods are well described in terms of technique and implementation (in Table 3a relative to Q1 and Table 3b relative to Q2, marked as D).The evaluations (observations and measurements) were well described, giving the diagnostic criteria used as well as information and instructions to observers (in Table 3a relative to Q1 and Table 3b relative to Q2, marked as E).The reproducibility of the research method was described for one observer (intra-observer action) and for several (minimum 3) observers (inter-observer action) (in Table 3a relative to Q1 and Table 3b relative to Q2, marked as F).The results are presented as relevant data needed for necessary calculations (in Table 3a relative to Q1 and Table 3b relative to Q2, marked as G).

#### 5.1.2. Moderate Level of Evidence

Research was assessed as having a moderate level of evidence if any of the above criteria were not met. On the other hand, a study with any of the deficits described below was assessed as having a low level of evidence.

#### 5.1.3. Low Level of Evidence

Research was judged to have a low level of evidence if it met any of the following criteria:The evaluation of the test and reference methods was independent (A).The population was not clearly described, and the spectrum of patients was distorted (B).The test method results influenced the decision to perform reference method (C).The test, reference method or both were not well described (D).The results were not well described (E).The reproducibility of the research method was not described or was only described for one observer (F).The results may have a systematic bias (H).The results were not presented in a way that enabled calculating effectiveness (G).

Quality assessments of the included research were performed using the risk-of-bias table in RevMan 5.3 for RCTs (Table 2a,b).

### 5.2. Evidence-Based Evaluation of Conclusions

The scientific evidence for the conclusions on diagnostic efficacy was considered strong, moderately strong, limited or insufficient depending on the quality and internal relevance (level of credibility) of the publications evaluated.

Strong research-based evidence: at least two publications or a systematic review must have a high level of evidence.Moderately strong research-based evidence: One publication must have a high level of evidence, and two subsequent publications must have a moderate level of evidence.Limited research-based evidence: at least two publications must have a moderate level of evidence.Insufficient research-based evidence: scientific evidence is insufficient or non-existent according to the criteria defined in this research.

Articles presented in the PRISMA flow diagram that obtained a minimum of 4 points were selected for final analysis (Table 2).

The results presented in the selected articles are summarised in Table 4a,b.

### 5.3. Evidence Synthesis

The results of this review are presented descriptively.

## 6. Results

This section was divided according to answers to Q1 and Q2.

### 6.1. Q1. New Cephalometric Analysis System in the Sagittal Plane

In the search, a total of 1451 records of articles were identified from the databases. In the first selection stage, 1046 articles were excluded, and another 74 duplicate items were removed. In the end, only 12 articles met the inclusion criteria outlined in the objectives.

### 6.2. Q2. Cephalometric Analysis Methods in the Horizontal Plane—The Evaluation of Vertical Defects

A total of 1451 records of articles were identified from the database search. In the first selection stage, 1046 articles were excluded. Only 5 articles were included for further analysis of the full text. At this stage, 1 more article was excluded as not meeting the criteria of the objectives. In the end, only 4 articles met the inclusion criteria outlined in the objectives.

## 7. Discussion of Outcomes

Since Broadbend’s and Hofrath’s introduction of the cephalometric analysis in 1931, many investigators have introduced further measurements and analyses to assess the skeletal or dentoalveolar basis of malocclusion [27,28]. A detailed cephalometric analysis is still an effective tool for the diagnosis and planning of orthodontic treatment. Unfortunately, like most additional methods of examination, the cephalometric analysis is not free of faults and errors, and thus, it should constitute a component of thoroughly conducted medical interviews and physical examinations to establish final diagnosis and implement the proper treatment of malocclusion. The basis for the systematic search for ever new analyses is the difficulty of mapping landmarks and the dependence of their position on growth and managed orthodontic treatment.

Nowadays, the increasing use and availability of CBCT equipment is largely related to the issue of radiological protection. CBCT is associated with higher radiation dose than OPG or cephalogram. However, if one projection provides the possibility of solving several diagnostic problems, it will replace registration from several projections in favour of one CBCT image. However, in order to be able to perform cephalometric analyses on the CBCT projection, it is necessary to evaluate the accuracy of introducing points such as n image and above all, to develop cephalometric analyses intended for such imaging. As long as such analyses and studies are created, it will be possible to refer them, among others, to the presented systematic review in order to compare the diagnostic value of 2D and 3D cephalometric analyses.

Cephalometric analysis addresses the origin of discrepancies in the sagittal and horizontal planes for the interrelationship of both jaws to identify anterior-–posterior and vertical malocclusions [1,2,3,4,5,6,7,8].

The measurements that are the gold standard in terms of the evaluation of sagittal relationships of maxillary bases include ANB angle, beta angle, Wits analysis, AF–-BF, MM–AB angle, AH–BH measurement and the Harvold index [7,23,25]. The main objections to the above-mentioned measurements include the instability of the S, N, Po, Or, A and B during growth, changes in their position during orthodontic treatment and the unreliability of their correct location on the cephalometric image that shows the patient’s lateral profile (a lateral cephalogram) [7,23,25]. Often, objections arise regarding the correct positioning of anthropometric points by various doctors and even by the same doctor in conducting subsequent analyses of the same patient at time intervals.

The current systematic review takes as its main objective the analysis of new angles and anthropometric measurements used in sagittal and horizontal analyses that were published after 2009. Most publications also discuss several previously used angular and linear measurements that are standards for individual cephalometric analyses, and, based on these analyses, they introduce new anthropometric points and cephalometric measurements [7,23,25].

The Yen angle determined by points S, M, G defines the sagittal relationship between the maxilla and mandible [10].

The measurement of the Yen angle was analysed in a published article by Neela P.K. et al. [10], Kumar and Sundareswaran [23] and Ali et al. [24].

Neela et al. [10] showed more stable sagittal analysis using the Yen angle compared with the previously used Wits, ANB and beta parameters. The elimination of the instability of the A and B points in the ANB analysis in relation to growth and changes due to orthodontic treatment, the functional occlusal plane in Wits and the condylar process axis in the beta angle analysis were considered to be the main advantages of Yen angle assessment.

The evidential value of the study by Neela et al. was found to be moderate, with its main shortcomings being the low number of participants in each group; the lack of comparative studies conducted at a time interval; and the lack of assessment of the stability, reliability and accuracy of the landmarks. The use of statistical analysis based on ANOVA was considered an advantage of the study. The shortcomings shown above indicate that to obtain high evidential value, studies should be conducted on larger groups, using comparative studies in the same study groups in periods before and after orthodontic treatment and by a group of investigators appropriately randomised to assess the reliability of the landmarks used in the analysis, in assessing both between-investigator and intra-observer significance.

Kumar and Sundareswaran [23] also found that the Yen angle is more reliable than the ANB angle, Wits appraisal and beta angle measurement, because it eliminates the difficulty of locating points A and B in ANB analysis, the functional occlusal plane in Wits appraisal and the axis of the condylar process in beta angle analysis. However, the authors point out a shortcoming in the analysis of the Yen angle: When there is rotation of the jaws, the actual sagittal discrepancy may be concealed. The evidential value of the study by Kumar and Sundareswaran was considered low as its main shortcoming was that it only systematised and described in chronological order the cephalometric analyses for assessing the sagittal jaw relationship available in the literature.

The shortcomings shown above indicate that to obtain a high evidential value, comparative studies should be performed for all parameters systematised in a systematic review. Studies should include large study groups, using comparative analysis in the same study groups in periods preceding and following the completion of orthodontic treatment, and be conducted by a group of investigators appropriately randomised to assess the reliability of the landmarks used in each analysis, in assessing both between-investigator and intra-observer significance. In large-scale studies of most cephalometric measurements used to assess sagittal discrepancy, standardised methodological criteria and comparisons of the significance of individual measurements using the same statistical analysis should be used for effective comparison.

Turker, Ozturk, Coban and Isgandarov [8] confirmed the validity of using Yen angle. In high-angle patients, Yen was found to be significantly lower than in low-angle patients and comparable with Pog–Nperp and Wits analysis in relation to beta angle, which was found to be significantly higher in this group. Beta and W angles were significantly lower in LA patients compared with HA patients. ANB, Beta, W and Yen angles showed significant correlations regardless of vertical facial growth type. The comparison of Yen angle analysis with other analyses in relation to the horizontal plane and vertical defects and the use of adjusted ANOVA were considered attributes of the presented study.

A comparison of only some parameters, without any indication of the selection criteria, was considered its fundamental shortcoming.

In order to obtain high evidential value, it would have been necessary to clarify the criteria for selecting the parameters to be assessed and to allow the comparison of the determinations of measurement values by different observers as well as by the same observer in comparative studies.

Ali et al. [24] showed reservations about the effectiveness of using the Yen angle in sagittal analysis. The authors showed that comparing beta, Yen and W angles with ANB angle and Wits appraisal does not show statistically significant differences in class III patients. A total of 100 lateral cephalograms were included in the study, and the use of only t-student statistical analysis was the shortcoming of this study. For this reason, the evidential value of the work was considered moderate. To increase the value of the evidence, it would have been necessary to use an advanced statistical method and determine whether the lack of statistically significant differences was only in Class III patients or whether the use of Yen angle analysis deviates from ANB, beta and W angle analyses and Wits appraisal in all skeletal classes determined in their comparison.

On the basis of the above studies, the use of Yen angle in the cephalometric analysis of sagittal discrepancy can be considered a valuable parameter because it is less dependent than the previously used ANB and beta angle and Wits appraisal to complement previous assessments.

### 7.1. Pi Analysis

Pi analysis was first taken into account by Kumar, Valiathan, Gautam, Chakravarthy and Jayaswal [12]. Pi analysis was also applied by Shetty, Desai, Kumar, Madhur and Alphonsa [26] and Kumar and Sundareswaran [23].

Kumar et al. [12] showed that the M and G landmarks used in Pi analysis are less susceptible to local changes associated with remodelling during growth or to secondary movements associated with remodelling during orthodontic treatment compared with A and B landmarks. The authors showed that the centroidality of the landmarks affects the precision of their determination and their invariability during growth, in contrast with previous standards such as A and B. The use of the true horizontal plane obtained in the NHP used in Pi analysis instead of another intracranial reference plane (SN, Frankfurt plane or occlusal, which have some specific limitations) results in the increased reliability of measurements. They showed that the comparison of NG’ with NM’ with normative values determines whether the defect originated from the maxilla or the mandible.

A limitation of Pi analysis is that it considers the position of the nasion during the growth period, which may change during actual jaw growth.

The evidential value of the study by Kumar et al. was found to be moderate. The main shortcomings of the above-mentioned study were the low number of participants in each group; the lack of comparative studies conducted at a time interval; and the lack of assessment of the stability, reliability and accuracy of the landmarks. The use of ANOVA was considered an advantage of the study. The demonstrated shortcomings indicate that for a high level of evidential value, studies should be conducted on larger groups using comparative analysis in the same study groups in the periods preceding and following the completion of orthodontic treatment and be conducted by a group of investigators appropriately randomised to assess the reliability of the landmarks used in each analysis, in assessing both between-investigator and intra-observer significance.

Pi analysis was included in a systematic review of 21 analyses for the assessment of anteroposterior discrepancy by Shetty et al. [26].

The evidential value of the study by Shetty et al. was considered low. The main shortcoming of the study was the systematic evaluation of only the individual parameters of sagittal discrepancy and not taking into account the criteria and evaluation of individual parameters on specific groups of subjects.

An advantage of the conducted study was the systematisation of known parameters for the assessment of sagittal discrepancy.

The demonstrated shortcomings indicate that for a high level of evidential value, studies should be conducted by randomised observers comparing the values of individual parameters for the assessment of maxillary base relationships in numerous groups of subjects before and after orthodontic treatment, assessing the rank of the suitability of individual parameters for the assessment of maxillary–mandibular relationships.

Kumar and Sundareswaran [23] also found that the Pi analysis is more dependable than ANB angle, Wits appraisal and beta angle because it eliminates the difficulty of locating points A and B in ANB analysis, the functional occlusal plane in Wits appraisal and the axis of the condylar process in beta angle analysis. In their discussion of cephalometric analyses in relation to Pi analysis, the authors stated that the highest level of correlation was obtained only for Pi angle and Pi linear (0.96)

The study presented here showed that Pi analysis, related to both the evaluation of Pi angle and the Pi linear measurement in determining the sagittal relationships of the maxilla and mandible, is a more objective analysis than the evaluation of the ANB angle, Wits appraisal or beta angle analysis.

### 7.2. Analysis of W Angle Determined by Points S, M, G Defining the Sagittal Relationship between the Maxilla and Mandible. W Angle Measured between the Line Perpendicular to Point M on SG Line and MG Line

W angle analysis was taken into account by Bhad, Nayak, Doshi [13], Kumar and Sundareswaran [23], Ali et al. [24], Shetty et al. [26] and Turker et al. [8].

Bhad et al. [13] also showed the higher stability of sagittal analysis using W angle compared with the previously used Wits, ANB and beta parameters. The elimination of the instability of the A, B and N points in the ANB analysis in relation to the growth and changes due to orthodontic treatment, the functional occlusal plane in Wits and the condylar process axis in beta angle analysis were considered to be the main advantages of W angle assessment. In the analysis of W angle, points S, G and M were used. By replacing the N point, which is unstable in the growth phase, with the S point, they proved that W angle measurement is more stable for the assessment of sagittal discrepancy than ANB angle, Wits analysis or beta angle.

The evidential value of the study by Bhad et al. was considered moderate, as the main shortcomings of this study were the low number of participants in each group; the limitation of the study to only a group that had not yet received orthodontic treatment without comparison with the stability of the results after the treatment; the lack of comparative studies at an interval; and the lack of assessment of the stability, reliability and accuracy of the landmarks. The use of ANOVA was considered an advantage of the study. 

Kumar and Sundareswaran [23] also found that W angle analysis is more reliable compared with ANB angle, Wits appraisal and beta angle because it eliminates the difficulty of locating points A, B and N in ANB analysis, the functional occlusal plane in Wits appraisal and the axis of the condylar process in beta angle analysis. In discussing cephalometric analyses in relation to W angle analysis, the authors noted a statistically significant difference between W angle measurement values in men and women for Class III diagnosis.

Ali et al. [24], in their comparative study, indicated that W angle analysis is the closest to ANB angle and Wits appraisal in Class I defects. In Class II and III defects, the authors questioned the effectiveness of assessing the sagittal relationship of the jaws based on W angle analysis. They claimed that it is less satisfactory compared with ANB, beta and Wits, which are considered standard. In their review, the authors considered the ANB angle to be the gold standard in the assessment of sagittal discrepancy.

The evidential value of the study by Ali et al. was considered moderate. Its main shortcomings were the small size of the study group, the limitation of the study to patients whose orthodontic treatment has not yet been initiated, the use of the t-student test as the only statistical tool, the lack of comparative studies and the lack of the comparative assessment of inter-observer and intra-observer stability of landmarks.

Shetty et al., in their systematic review of 21 studies assessing anteroposterior discrepancy, confirmed the value of using W angle analysis [26].

The study presented here showed that the analysis of W angle in determining the sagittal relationship between the maxilla and mandible is more objective than the assessment of the ANB, Wits or beta angle, although ANB, Wits and beta are considered by some authors to be the gold standard in the assessment of the sagittal relationship.

### 7.3. SAR Angle

SAR angle measurement was analysed in a study by Sonahita et al. [14], Kumar and Sundareswaran [23] and Shetty S. et al. [26].

Sonahita et al. [14] proved that M, G and W used in the analysis are more stable and reliable as they are easier to find compared with A, B and N and do not undergo changes due to growth and transformations associated with orthodontic treatment. Therefore, SAR analysis was demonstrated to have higher reliability than ANB, beta and Wits analyses.

The evidential value of the study by Sonahita et al. was found to be moderate since the main shortcomings of the above-mentioned study were the low number of participants in each group; the lack of comparative studies conducted at a time interval; and the lack of assessment of the stability, reliability and accuracy of the landmarks. The use of ANOVA was considered an advantage of the study. The demonstrated shortcomings indicate that for a high level of evidential value, studies should be conducted on larger groups, using comparative analysis in the same study groups in periods preceding and following the completion of orthodontic treatment and be conducted by a group of investigators appropriately randomised to assess the reliability of the landmarks used in each analysis in assessing both between-investigator and intra-observer significance.

Kumar and Sundareswaran [23] also found that SAR angle analysis is more reliable than ANB, Wits and beta because it eliminates the difficulty of finding A, B and N points in ANB analysis. In their conclusion, the authors emphasised that the rotational effects of the jaws; the variable positions of A, B and N; changes in the length of the skull base; tooth eruption; the curve of Spee, etc. seem to affect the assessment of the position of the mandible in relation to the maxilla, which also results in the use of extracranial reference planes. At the same time, using only one of the cephalometric analyses may not provide a correct diagnosis. Therefore, for a correct diagnosis, several of the angular or linear parameters should be used without considering only one type of measurement as being the only valid one in assessing the relationship to the jaw.

Shetty et al. [26], in their systematic review of 21 studies assessing anteroposterior discrepancy confirmed the value of using SAR angle analysis [26] The authors came to similar conclusions as Kumar and Sundareswaran. In their conclusion, they emphasized that rotational effects of the jaws, variable positions of points A and B, nasion, changes in skull base length, tooth eruption, curve of Spee etc., seem to affect the assessment of the position of the mandible in relation to the maxilla which also results in the use of extracranial reference planes. At the same time, using only one of the cephalometric analyses may not provide a correct diagnosis. Therefore, for a correct diagnosis, several of the angular or linear parameters should be used without considering only one type of measurement as being the only valid one in assessing the relationship to the jaw.

In their conclusion, the authors emphasised that the rotational effects of the jaws, the variable positions of points A, B and nasion, changes in the length of the skull base, tooth eruption, the curve of Spee, etc., seem to affect the assessment of the position of the mandible in relation to the maxilla which also results in the use of extracranial reference planes. At the same time, using only one of the cephalometric analyses may not provide a correct diagnosis. Therefore, for a correct diagnosis, several of the angular or linear parameters should be used without considering only one type of measurement as being the only valid one in assessing the relationship to the jaw.

Based on the study presented here, the value of SAR angle analysis as a parameter helpful in assessing sagittal discrepancy and the position of the maxillary bases should be recognised.

### 7.4. DW Angle Using Walker’s and Wing (WW) Point

DW plane measurement has been analysed in a study by Hatewar et al. [15], Kumar and Sundareswaran [23] and Shetty et al. [26].

Hatewar et al. [15] showed significant regularity in the evaluation of the sagittal relationships of jaw bases. The measurements so far considered to be authoritatively established for the assessment of the sagittal relationship often turn out to be inaccurate because they are based on various angular and linear measurements related to the position of N, A and B points. The landmarks used in the DW analysis are characterised by higher stability and repeatability, reliability and invariability in relation to growth processes and changes resulting from orthodontic treatment, in contrast to N, A and B, which increases the unambiguity of the measurements. In DW plane analysis, linear measurements are performed to determine the vertical mandibular dimension, and angular measurements are taken to determine the rotation of the jaw.

The evidential value of the study by Hatewar et al. was found to be moderate as the major drawbacks of this study include the low number of participants in each group; limitation of the study to one race without interracial comparison; no comparison studies made at time intervals; no assessment of the stability, reliability and accuracy of landmarks; ad the use of Student’s *t*-test statistical method only. The use of the analysis in different age groups from 8 to 27 years was considered an advantage for the study.

The demonstrated shortcomings indicate that for a high level of evidential value, studies should be conducted on larger groups, using comparative analysis in the same study groups in periods preceding and following the completion of orthodontic treatment and be conducted by a group of investigators appropriately randomised to assess the reliability of the landmarks used in each analysis, in assessing both between-investigator and intra-observer significance. Moreover, an extensive statistical analysis based on ANOVA should be applied.

The evidential assessment makes it possible to conclude that measurements relative to the DW plane are more effective compared with the ANB, Wits or beta measurements that are routinely used for assessing sagittal discrepancy.

### 7.5. Tau Angle

The measurement of Tau angle was analysed by Gupta, Singh, Tripathi, Gopal, and Rai [6].

The Tau angle determined by points T, M, G and defining the sagittal relationship between the maxilla and mandible.

The authors found higher precision and reliability in the assessment of jaw sagittal relationships by using points G and M, with higher invariance than A and B, and Tau point that is more reliable than S compared to routinely used ANB and Beta angle analyses and Wits analysis. The authors indicate that the obtained results define a skeletal ratio that depends on stable landmarks. At the same time, they reveal that the assessment is not affected by jaw rotation in the vertical dimension due to growth or implemented orthodontic treatment.

The evidential value of the study by Gupta et al. was considered moderate. The main shortcomings of the study include the inequality of various groups of participants; the large age range of the patients studied in one group that included both adolescent and adult patients; the lack of comparative studies before and after orthodontic treatment; the lack of comparative studies made at time interval; and the lack of an assessment of stability, reliability and accuracy of landmarks. The use of ANOVA was considered an advantage of the study.

The demonstrated shortcomings indicate that for a high level of evidential value, studies should be conducted on larger age-equivalent groups, using comparative analysis in the same study groups in periods preceding and following the completion of orthodontic treatment, and be conducted by a group of investigators appropriately randomised to assess the reliability of the landmarks used in each analysis, in assessing both between-investigator and intra-observer significance. It seems necessary to confirm the dependence of evaluated parameters on rotation of the mandible and maxilla that are related to the patient age.

Only one study does not enable firm conclusions supporting the reliability of the Tau angle analysis in assessing sagittal discrepancy and requires further research.

### 7.6. R Angle

The R angle was measured by Rizwan, Mascarenhas [16], Ahmed, Shaikh and Fida [25]. It is determined by points N, C and Me [16].

Rizwan et al. [16] proved that the R angle is constructed from minimal cephalometric landmarks that can be easily and accurately located on digital lateral cephalograms. Over them, the R angle is constructed using only fixed skeletal landmarks and no constructed points or landmarks, thereby minimising operator error. The C–N axis and C–Me axis are more stable compared with the currently used unstable planes.

A systematic description of various cephalometric measurements compared with vertical defects is favourable for the study. Attention should be paid to a chronological review of methods for assessing vertical discrepancy and a discussion concerning the reliability and shortcomings of various anthropometric points, lines and planes, as well as the skewed values of parameters used for assessing vertical defects.

The evidential value of the study by Rizwan et al. was considered moderate. The main shortcomings of the study include the low number of participants; the specified and limited ages of the participants (18–27 years); the lack of comparative studies before and after the orthodontic treatment; the lack of comparative studies made at time intervals; and the lack of the assessment of stability, reliability and accuracy of landmarks. The use of statistical analysis based on ANOVA was considered an advantage of the study.

The demonstrated shortcomings indicate that for a high level of evidential value, studies should be conducted on larger age-equivalent groups, using comparative analysis in the same study groups in periods preceding and following the completion of orthodontic treatment, and be conducted by a group of investigators appropriately randomised to assess the reliability of the landmarks used in each analysis, in assessing both between-investigator and intra-observer significance. There should also be comparisons of the reliability of the lines that represent the size of the angle at different development stages of patients from selected high-, medium- and low-angle skeletal groups.

A comparative evaluation of the R angle was also made by Ahmed, Shaikh and Fida [25].

The authors denied the value of the R analysis in assessing vertical discrepancy. In the light of obtained results, the authors proved that SN.GoGn and FMA are the most reliable indicators, while LAFH.TAFH turned out to be the least reliable indicator in assessing the vertical facial growth pattern.

Given that the article discusses the results of twice as many participants in each vertical discrepancy group, its evidential value can be considered higher than the results of the study by Rizwan M. et al.

The evidential value of the study by Ahmed et al. was considered moderate. The main shortcomings of the study include the low number of participants; the lack of comparative studies before and after orthodontic treatment; the lack of comparative studies made at time intervals; and the lack of the assessment of stability, reliability and accuracy of landmarks. The use of statistical analysis was considered an advantage of the study. Kappa statistics were used for comparing the diagnostic accuracy of different analyses. To further validate the results, sensitivity and positive predictive value (PPV) were calculated for each parameter.

The demonstrated shortcomings indicate that for a high level of evidential value, studies should be conducted on larger age-equivalent groups, using comparative analysis in the same study groups in periods preceding and following the completion of orthodontic treatment and be conducted by a group of investigators appropriately randomised to assess the reliability of the landmarks used in each analysis in assessing both between-investigator and intra-observer significance. There should also be comparisons of the reliability of the lines that represent the size of the angle at different development stages of patients from selected hyperdivergent, normodivergent and hypodivergent skeletal groups.

The conflicting results of published studies need repeated and more detailed research to make constructive conclusions about the usefulness of R angle analysis in assessing vertical relationships.

### 7.7. KP Plane (Extraoral)

The measurement of the KP plane was analysed by Kattan, Kattan and Elhiny [17].

An innovation in the creation of plane K is the use of an extraoral orientor when taking a 2D cephalogram to determine the natural head position (NHP).

The authors found a high correlation with the Frankfurt plane; however, due to the higher reliability and accuracy of the determination of the new plane K, the obtained results of the vertical discrepancy measurements give results that are similar to real values, minimising the risk of error that is associated with the determination of Po and Or points.

The correct assessment of plane K was dependent on NHP and the positioning of the orientor to stabilise the head in its natural position.

The authors noted that the suggested position can be used for two-dimensional radiography and computed tomography.

The authors’ conclusion is that determination of the plane K compared to the Frankfurt plane is much more reproducible and accurate in cephalometric analysis, especially when using an orientor for head orientation in NHD. The introduction of plane K gives an advantage over previous analyses by introducing an additional stable orientor for the correct orientation of the patient’s head in NHP.

The evidential value of the study by Kattan et al. was considered moderate. The main shortcomings of the study include the low number of participants; the lack of comparative studies before and after orthodontic treatment; the lack of comparative studies made at time intervals; and the lack of the assessment of stability, reliability and accuracy of landmarks, as well as the lack of unbiased statistical analysis.

The demonstrated shortcomings indicate that for a high level of evidential value, studies should be conducted on larger age-equivalent groups, using prospective, retrospective and comparative studies in the same study groups in periods preceding and following the completion of orthodontic treatment and be conducted by a group of investigators appropriately randomised to assess the reliability of the landmarks used in each analysis in assessing both between-investigator and intra-observer significance. It would be advisable to study a larger number of groups and to compare results at different development periods.

A further analysis and additional tests are necessary to determine diagnostic values of plane K.

### 7.8. The Superior Border of Zygomatic Arch

The effectiveness of the use of a plane A, the superior border of the zygomatic arch, was proved by Park, Lee, Koh and Song W.C. [18].

The study was developed using 3D cone–beam computed tomography. A 3D–2D transcription seems possible, although this would require further research.

The authors revealed that the proposed new plane of the zygomatic arch may be an alternative plane to the Frankfurt line but more reliable for the assessment of vertical discrepancy.

The applied analytical and statistical methods proved the stability of the measurement between the Frankfurt line and the zygomatic arch line.

The evidential value of the study by Park et al. was considered moderate. The main shortcomings of the study include the low number of participants, limiting the age of participants to 21–30 years, the use of only 3D examinations, the lack of comparative studies before and after orthodontic treatment; the lack of comparative studies made at time intervals; and the lack of the assessment of stability, reliability and accuracy of landmarks, as well as the lack of unbiased statistical analysis, which was limited to the Student’s *t*-test analysis.

The demonstrated shortcomings indicate that to obtain a high degree of evidential value, 2D examinations should be conducted, groups should be larger and age-equivalent and examinations should be conducted in different age groups including groups of patients in the developmental period. Moreover, comparative studies should be carried out in the same study groups in periods before and after orthodontic treatment.

A further analysis and additional tests are necessary to determine diagnostic values of the plane of the zygomatic arch.

## 8. Conclusions

The evidence suggests that there are many new reference points and cephalometric indices that can be successfully used for determining the sagittal discrepancy in the mutual position of the maxillary bases. However, although the systematic review has low heterogeneity, the included studies exhibited a moderate risk of bias and low to moderate quality. Future studies are required with adequate internal and external validity. Sagittal discrepancy and assessment methods of the relationship between the upper jaw and lower jaw should be more accurate in the future.

In terms of the new cephalometric measurements to determine discrepancy in the vertical dimension, the number of performed studies is limited, with very low quality and moderate risk of bias of the study assessed. Simultaneously, the review revealed the existence of novel alternative parameters to contemporary measurements in 3D examinations. Tracing the process of determining reference points, lines and anthropometric planes in volumetric tomography also seems possible in 2D. However, such suggestions require further research and analysis. Current studies do not seem to be very constructive.

Due to the radiological protection of patients and the tendency to limit exposure to X-rays, it seems necessary to use 2D cephalometric diagnosis and, only in borderline cases, 3D diagnosis.

## Figures and Tables

**Figure 1 diagnostics-12-01723-f001:**
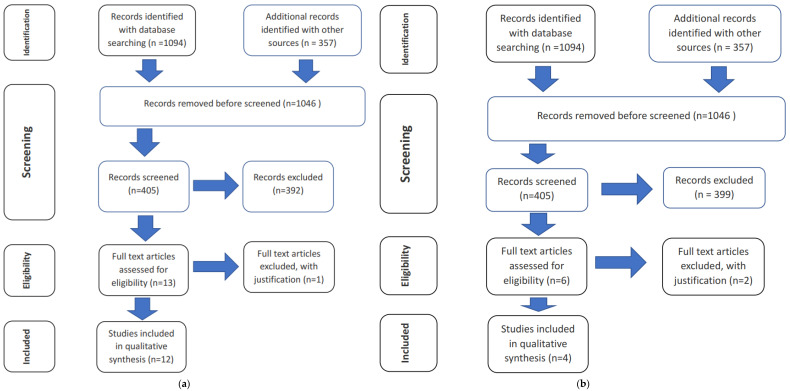
(**a**) The methodology used to select articles in relation to sagittal analysis. (**b**) The methodology used to select articles in relation to horizontal analysis [8].

**Table 1 diagnostics-12-01723-t001:** The algorithms used in the search strategy updated for each database and question.

Pico strategy	Population: Patients with orthodontic treatment Interventions: cephalometric 2D Comparator: Q1 = sagittal analyses, Q2 = horizontal analyses Outcomes: new indicator of sagittal dysplasia: YEN angle, W angle, Pi angle, Tau angle, SAR angle, ODI, APDI, HBN angle, DW plane, AF–BF, Another analysis ANB Angle, Wits marker, ROC, beta angle, Downs angle, AB plane angle
Focused questions	Q1 = Wich is the effect on the new landmarks and measurements in the cephalometric analyses vs. conventional analyses of the sagittal relationships of the jaws Q2 = Wich is the effect on the new landmarks and measurements in the cephalometric analyses vs. conventional analyses of the horizontal relationships of the jaws
Number of registers found for each database	Algorithms used in the search strategy adapted for each database and question
PubMed Q1 = 1451 (12) Q2 = 1451 (8)	Q1 = Cephalometr* and (orthodontic* or ‘orthodontic treatment planning’) and (‘efficacy’ or ‘reproducibility’ or ‘repeatability’ or ‘reliability’ or ‘accuracy’ or ‘validity’ or ‘validation’ or ‘precision’ or ‘variability’ or ‘efficiency’ or ‘comparison’) and (YEN Angle or W Angle or Pi Angle or Tau Angle or SAR Angle or ANB Angle or Wits marker or ODI or APDI or ROC or Beta Angle or Downs Angle or AB plane Angle or HBN Angle or DW plane or AF-BF) not (‘Cone-Beam Computed Tomography’ or ‘Three-Dimensional imaging’ or ‘Cone Beam Computed Tomography’ or ‘Cone Beam CT’ or ‘Volumetric Computed Tomography’ or ‘Volume Computed Tomography’ or ‘Volume CT’ or ‘Volumetric CT’ or ‘Cone beam CT’ or ‘CBCT’ or ‘digital volume tomography’ or ‘DVT’ or ‘Spiral Computed Tomography’ or ‘Spiral Computer-Assisted Tomography’ or ‘Spiral Computerized Tomography’ or ‘spiral CT Scan’ or ‘spiral CT Scans’ or ‘Helical CT’ or ‘Helical CTS’ or ‘Helical Computed Tomography’ or ‘Spiral CAT Scan’ or ‘Spiral CAT Scans’ or ‘3D’ or ‘3-D’ or ‘three dimension*’).) AND ((“2013/01/01”[Date—Completion]: “3000”[Date—Completion])) Q2 = Cephalometr* and (orthodontic* or ‘orthodontic treatment planning’) and (‘efficacy’ or ‘reproducibility’ or ‘repeatability’ or ‘reliability’ or ‘accuracy’ or ‘validity’ or ‘validation’ or ‘precision’ or ‘variability’ or ‘efficiency’ or ‘comparison’) and (ODI or DW plane or zygomatic arch or foramina of the trigeminal nerve landmarks or Frankfurt line or orbito-ingotic line or gonial angle or AF-BF) not (‘Cone-Beam Computed Tomography’ or ‘Three-Dimensional imaging’ or ‘Cone Beam Computed Tomography’ or ‘Cone Beam CT’ or ‘Volumetric Computed Tomography’ or ‘Volume Computed Tomography’ or ‘Volume CT’ or ‘Volumetric CT’ or ‘Cone beam CT’ or ‘CBCT’ or ‘digital volume tomography’ or ‘DVT’ or ‘Spiral Computed Tomography’ or ‘Spiral Computer-Assisted Tomography’ or ‘Spiral Computerized Tomography’ or ‘spiral CT Scan’ or ‘spiral CT Scans’ or ‘Helical CT’ or ‘Helical CTS’ or ‘Helical Computed Tomography’ or ‘Spiral CAT Scan’ or ‘Spiral CAT Scans’ or ‘3D’ or ‘3-D’ or ‘three dimension*’).) AND ((“2013/01/01”[Date—Completion]: “3000”[Date—Completion]))
Google Scholar Q1 = 7 (1) Q2 = 0	Q1 = Cephalometr * and (orthodontic * or ‘orthodontic treatment planning’) and (‘efficacy’ or ‘reproducibility’ or ‘repeatability’ or ‘reliability’ or ‘accuracy’ or ‘validity’ or ‘validation’ or ‘precision’ or ‘variability’ or ‘efficiency’ or ‘comparison’) and (YEN Angle or W Angle or Pi Angle or Tau Angle or SAR Angle or ANB Angle or Wits marker or ODI or APDI or ROC or Beta Angle or Downs Angle or AB plane Angle or HBN Angle or DW plane or AF-BF) not (‘Cone-Beam Computed Tomography’ or ‘Three-Dimensional imaging’ or ‘Cone Beam Computed Tomography’ or ‘Cone Beam CT’ or ‘Volumetric Computed Tomography’ or ‘Volume Computed Tomography’ or ‘Volume CT’ or ‘Volumetric CT’ or ‘Cone beam CT’ or ‘CBCT’ or ‘digital volume tomography’ or ‘DVT’ or ‘Spiral Computed Tomography’ or ‘Spiral Computer-Assisted Tomography’ or ‘Spiral Computerized Tomography’ or ‘spiral CT Scan’ or ‘spiral CT Scans’ or ‘Helical CT’ or ‘Helical CTS’ or ‘Helical Computed Tomography’ or ‘Spiral CAT Scan’ or ‘Spiral CAT Scans’ or ‘3D’ or ‘3-D’ or ‘three dimension*’).) AND ((“2013/01/01”[Date—Completion]: “3000”[Date—Completion])) Q2 = Cephalometr* and (orthodontic* or ‘orthodontic treatment planning’) and (‘efficacy’ or ‘reproducibility’ or ‘repeatability’ or ‘reliability’ or ‘accuracy’ or ‘validity’ or ‘validation’ or ‘precision’ or ‘variability’ or ‘efficiency’ or ‘comparison’) and (ODI or DW plane or zygomatic arch or foramina of the trigeminal nerve landamrks or francfort line or orbito -ingotic line or gonial angle or AF-BF) not (‘Cone-Beam Computed Tomography’ or ‘Three-Dimensional imaging’ or ‘Cone Beam Computed Tomography’ or ‘Cone Beam CT’ or ‘Volumetric Computed Tomography’ or ‘Volume Computed Tomography’ or ‘Volume CT’ or ‘Volumetric CT’ or ‘Cone beam CT’ or ‘CBCT’ or ‘digital volume tomography’ or ‘DVT’ or ‘Spiral Computed Tomography’ or ‘Spiral Computer-Assisted Tomography’ or ‘Spiral Computerized Tomography’ or ‘spiral CT Scan’ or ‘spiral CT Scans’ or ‘Helical CT’ or ‘Helical CTS’ or ‘Helical Computed Tomography’ or ‘Spiral CAT Scan’ or ‘Spiral CAT Scans’ or ‘3D’ or ‘3-D’ or ‘three dimension*’).) AND ((“2013/01/01”[Date—Completion]: “3000”[Date—Completion]))
Pro Quest Q1 = 112 (2)	Cephalometr* and (orthodontic* or ‘orthodontic analysis’) and (2D lateral cephalometry) and (W angle or YEN angle or Pi ANgle or Tau Angle)
Web of Science Q1 = 1 Q2 = 0	Q1 = Cephalometr and (orthodontic or ‘orthodontic analysis) Q2 = Cephalometr and (orthodontic or ‘orthodontic analysis)

**Table 3 diagnostics-12-01723-t003:** (a) The evaluation of the conclusions according to the degree of evidence of articles discussing new indicators of cephalometric analysis in relation to the sagittal plane. (b) The evaluation of the conclusions by degree of evidence of articles discussing new indicators of cephalometric analysis in relation to the horizontal plane.

**(a)**
**Q1**	**Author (year)**	**Neela 2009** [10]	**Bhad 2011** [13]	**Kumar 2012** [12]	**Kumar 2014** [23]	**Sonahita 2014** [14]	**Hatewar 2015** [15]	**Ali 2018** [24]	**Ahmed 2018** [25]	**Shetty****2019** [26]	**Gupta 2020** [6]	**Jedliński 2020** [7]	**Gokhan 2021** [8]
Level of evidence												
**(b)**
**Q2**	**Author (year)**	**Rizwan 2013** [16]	**Ahmed M. 2016** [25]	**Kattan EE.****2018** [17]	**Park JA. 2019** [18]
Level of evidence				

Yellow—moderate level of evidence, Green—low level of evidence.

**Table 4 diagnostics-12-01723-t004:** (**a**) Publication analysis of parameters in sagittal discrepancy analysis. (**b**) Publication analysis of parameters in vertical discrepancy analysis.

**(a)**									
	**Authors (Year)**	**Title**	**Aim of the Study**	**Observers**	**Studium Project**		**Statistical Method**	**Results According to Authors**	**Level of Evidence**
2009	Neela PK, Mascarenhas R, Husain A. [10]	A new sagittal dysplasia indicator: the YEN angle.	The development of a new cephalometric measurement to assess the sagittal relationship between maxilla and mandible. YEN angle.	75 lateral cephalograms before treatment (25 each in classes I, II and III)	The new measurement is based on landmarks S, M (midpoint of the anterior maxilla) and G (centre mandibular symphysis). YEN angle measured in M.		The mean and standard deviation for YEN angle were calculated in all three skeletal groups. One-way analysis of variance (ANOVA) and Newman–Keuls test were used.	Aim: to improve the reliable assessment of sagittal relationship between the two jaws. 117° < YEN < 123° skeletal class I. With YEN < 117° skeleton class II YEN > 123° skeletal class III.	moderate
2011	Bhad WA, Nayak S, Doshi UH. [13]	A new approach to the assessment of sagittal dysplasia: the W angle.	The development of a new cephalometric measurement to assess the sagittal relationship between maxilla and mandible. W angle.	142 cephalometric radiographs before treatment of patients aged 15 to 25 years.	The new measurement is based on landmarks S, M (midpoint of the anterior maxilla) and G (centre mandibular symphysis) and W angle measured between the perpendicular from point M on the S–G line and on the M–G line.		Mean and standard deviation for W angle were calculated. One-way analysis of variance and Newman–Keuls test were applied	51° < W < 56° skeletal class I. W < 51° degrees skeletal class II. W > 56° degrees skeleton class III.	moderate
2012	Kumar S, Valiathan A, Gautam P, Chakravarthy K, Jayaswal P. [12]	An evaluation of the Pi analysis in the assessment of the anteroposterior jaw relationship.	The development of a new cephalometric measurement to assess the sagittal relationship between maxilla and mandible. Pi angle and the linear value of Pi.	155 persons average age 19.7 years	The trial was divided into class I, II or III skeletal groups based on the ANB angle. Descriptive data were calculated for each variable and group.		The correlation coefficients between class I parameters were calculated. Coefficient of determination, regression coefficient, regression equation, standard error of estimation.	$\bar{Pi}=$ 3.40 (±2.04) class I $\bar{Pi}=$ 8.94 (±3.16) class II $\bar{Pi}=$ 3.57 (±1.61) class III For linear Pi = 3.40 (±2.20) class I, Pi = 8.90 (±3.56) class II Pi = 3.30 (±2.30) class III Pi angle > 5°; 89% sensitivity, 82% specificity in distinguishing class II skeletal group from class I. Pi angle < 1.3°; 100% sensitivity, 84% specificity in distinguishing class III skeletal groups from class I. The accuracy of distinguishing class II groups from class I was = 85% and that of class III from class I = 90%. The cut-off point between classes I and II may be regarded as the angle Pi = 5° between classes I and III, Pi = 1.3° No correlation Pi-ANB Pi-Beta, Pi-WITS The highest level of correlation was obtained for angle Pi and linear Pi (0.96).	moderate
2014	Kumar V., Sumdareswaran S., [23]	Cephalometric Assessment of Sagittal Dysplasia: A Review of Twenty-One Methods	The review provides an insight into the various cephalometric methods used to assess the sagittal relationships of jaws in chronological order and their implications in modern orthodontics.	21 analyses of the sagittal plane	Fixed values for linear measurements were discussed Glenoid fossa–sella Sella–Ptm Maxillary lengh Ptm to upper 6 Mandibular length of angle measurements: angle between NPog and AB line Angle of convexity NA to APog ANB angle Tailor`s AB” distance B orthogonal projection on SN of the line and orthogonal to this line drawn from A AXD angle and AD distance Wits APDI angle AXB angle JYD angle Maxillo–mandibular difference calculated as the angle between A–Co and Gn–Co AF–BF distance (distance between projections A and B on the Frankfurt plane Quadrilateral analysis between SN–PP–G–TG and NAG angle APP–BPP distance as a distance between projections A and B on PP (plane of the jaw base ANS–PNS FABA analyses of the angles of AB to FH and AB to the parallel shift FH through A Beta angle formed by Co–B, AB plane and the orthogonal to Co–B descending from A Yen–SMG angle		none	Details of 21 measurements to determine maxilla and mandible sagittal position	low
2015	Sonahita A.; Jitendra B., Praveen M., Sudhir K., Kumar JR [14]	The SAR Angle: A Contemorary Sagital Jaw Dysplasia Marker.	The aim is to determine means and standard deviation for this angle in persons with skeletal classes I, II and III.	60 pretreatment lateral cephalograms of 13–25 years old patients	SAR angle is a new parameter for assessing apical base sagittal discrepancy. It uses three skeletal reference points: Point M: Midpoint of the premaxilla Point G: Centre of the largest circle that is tangent to the internal inferior, anterior and posterior surfaces of the mandibular symphysis Point W (Walker’s point): The mean intersection point of the lower contours of the anterior clinoid processes (ACP) and the contour of the anterior wall of the sella turcica. The three lines that would form joining these points include • the line connecting Point M and Point G • the line connecting Point W and Point G • and the line from point M perpendicular to the W–G line. The angle to measure is between the perpendicular line from point M to W–G, while the M–G line is the SAR angle		The data were summarized as mean ± SD. Groups were compared by factor analysis (gender and class), analysis of variance and Newman–Keuls post hoc test. Receiver operating characteristics (ROC) curve analysis was performed to evaluate the sensitivity and specificity of SAR angle as a differential test between the three skeletal groups.	The mean SAR angle = 55.98° (SD 2.24), Class I skeletal pattern group SAR angle = 50.18° (SD 2.70) Class II SAR angle = 63.65°(SD 2.25) Class III skeletal group 53° < SAR < 59° Class I skeletal pattern; SAR < 53° Class II skeletal pattern SAR > 59° Class III skeletal pattern.	moderate
2015	Hatewar SK., Reddy GH., Singh JR., Jain M., Munje S., Khandelwal P. [15]	A new dimension to cephalometry: DW plane. The access to the skeletal jaw discrepancy using Walkers point.	This study aims to establish a new cephalometric measurement to assess skeletal jaw discrepancy using Walker’s point.	100 lateral cephalograms of indigenous peoples of the Americas aged 8–10, 12–18, 19–27 years.	Point A, Point B, Walker’s point (W) and wing point (w) were used for indicating the severity and type of skeletal dysplasia. Double W (DW) was constructed joining the Walker’s and wing points.		The analysis of variance and Student’s *t*-test were applied, which revealed significant results.	The DW plane is an effective way to accurately establish skeletal jaw relationships. It analyses the variance between linear measurements to determine the sagittal jaw relationship, linear measurements for vertical maxillary height and angular measurements to determine rotational jaw changes. This linear difference of 8.2 ± 0.9 mm indicated a Class I skeletal pattern.	low
2018	Ali SM., Manjunath G., Sheetal A. [24]	A Comparison of 3 New Cephalometric Angles with ANB and Wits Appraisal for Assessing Sagittal Jaw Relationship	To study the comparison of ANB and Wits appraisal with 3 new cephalometric angles.	100 lateral cephalometric radiographs	ANB angle evaluation, Wits evaluation, beta angle, AB plane angle, YEN angle and W angle.		Student’s *t*-test	Student’s *t*-test showed, in Class I = 100%, correlation with ANB. The closest angle was W angle when compared with ANB and Wits appraisal. In the Class II samples, beta angle was closest compared with ANB, whereas Yen and W angles showed considerable differences in comparison with ANB and Wits appraisal. The comparisons of beta, Yen, and W angles with ANB angle and Wits appraisal in Class III samples revealed no significant differences. The statistical comparison of the overall mean beta, yen, and W angles was 1, 0.53, 0.47, and 0.53, respectively, for Classes I, II, and III samples with ANB and Wits = 100% correlation compared with ANB and Wits appraisal. There is no gold standard for ANB angle. Beta, Yen, and W angles are not accurate r consistent, showing varying results fors classes I-III compared with ANB.	moderate
2018	Ahmed M, Shaikh A, Fida M. [25]	Diagnostic validity of different cephalometric analyses for assessment of the sagittal skeletal pattern.	Reliability and relevance assessment of various skeletal analyses to identify sagittal skeletal pattern.	146 persons (men = 77; women = 69; mean age = 23.6 ± 4.6 years).	The assessment of the anteroposterior skeletal system using: ANB angle, Wits, Beta angle, angle of the AB plane, Downs convexity angle, W angle.		The accuracy and reliability of the above analyses were determined using the Kappa statistic, sensitivity and positive predictive value (PPV).	ANB highest diagnostic agreement (k = 0.802). In the class I group, Downs convex angle showed the highest sensitivity (0.968), and ANB showed the highest PPV (0.910). In the class II group, ANB angle (0.928) and PPV (0.951) showed the highest sensitivity. In the class III group, ANB angle, Wits appraisal and Beta angle showed sensitivity (0.902). Downs convex angle and ANB angle showed the highest sensitivity (1.00). Conclusion: the ANB angle was found to be the most relevant and reliable indicator in all sagittal groups. Downs angle, Wits appraisal and Beta angle can be used as valid indicators to assess class III sagittal pattern.	moderate
2019	Shetty SK., Desai SJ., Kumar M., Madhur VK., Alphonsa BM., [26]	Cephalometric Assessment of Anterioposterior Discrepancy: A Review of Various Analyses in Chronological Order	Previously established parameters like: ANB angle, Wits, AF-BF, APDI, Beta angle, Yen angle, W angle, Pi analysis, SAR angle, HBN angle, DW plane Chronologic order and its clinical implications in contemporary orthodontics.	21 analyses	Previously, a total of 21 cephalometric analyses were performed to determine the anteroposterior position of the mandible in the sagittal plane.		none	The rotational effects of jaws, variable positions of points A and B, nasion, variations in cranial base length, tooth eruption, curve of Spee, etc. appear to influence anteroposterior assessment, resulting in the employment of extracranial reference planes as well. One cephalometric analysis may not result in an accurate diagnosis. Moreover, cephalometry is not a specific science or method, and therefore numerous analyses supported by angular and linear parameters have obvious limitations.	low
2020	Gupta P, Singh N, Tripathi T, Gopal R, Rai P. [6]	Tau Angle: new approach to assessing true sagittal skeletal maxillomandibular relationship.	Present new Tau angle used in cephalometric analysis.	Age group of 13- to 30-year-olds. Class I consisted of 101 patients (51 males, 50 females). Class II consisted of 101 patients (51 males, 50 females). Class III consisted of 77 patients (37 males, 40 females).	Tau angle is a novel parameter for determining the true bony sagittal maxillomandibular relationship. Tau angle is constructed by marking three cephalometric landmarks: Point T: The uppermost point at the junction of the frontal wall of the pituitary fossa and tuberculum sellae; Point M: The constructed point representing the centre of the biggest circle that is tangent to the frontal, upper and palatal surfaces of the maxilla; Point G: The focal point of the biggest circle that is tangent to the inner frontal, posterior and lower edges of the mandibular symphysis. Tau angle lies between the two lines connecting T and G points as well as M and G points. This study aims to establish Tau angle’s mean and standard deviation for three skeletal malocclusions.		The normality of the data was assessed by skewness, kurtosis and Shapiro–Wilk test. ANOVA and Dunnett’s T3 post hoc test determine differences among the three skeletal patterns. Student’s *t*-test	The mean and standard deviation for Tau angles in the class I, II, and III groups were 31.93 (±1.68)°, 38.32 (±2.93)° and 25.54 (±2.85)°, respectively. The ANOVA and Dunnett’s T3 test revealed significant differences in the mean Tau angle among three groups (*p* ≤ 0.05). *T* tests conveyed no significant difference in terms of Tau angle values between sexes in each skeletal pattern. Tau angle at 34.25° is 96% sensitive and 98% specific in differentiating class II and I. Therefore, ROC curves set the Tau angle cut-off points of class III and II skeletal patterns with class I to be approximately 28.5° and 34.25°, respectively.	moderate
2020	Jedliński M., Janiszewska-Olszowska J., Grocholewicz K., [7]	Description of the sagittal jaw relation in cephalometric analysis—a review of literature	present the most frequently used cephalometric measurements to assess the skeletal class on a lateral cephalometric headfilm		ANB angle, WITS appraisal, APDI Harvold analysis		none	ANB angle cannot be used as the only indicator of sagittal skeletal discrepancy. WITS appraisal is independent of the variability of cranial base structures and thus may be an important supplement to the diagnosis, although it depends on the variability of the occlusal plane. APDI can reliably distinguish between class I, II and III malocclusion.	low
2021	Turker G, Ozturk T, Coban G, Isgandarov E. [8]	Evaluation of Various Sagittal Cephalometric Measurements in Skeletal Class I Individuals with Different Vertical Facial Growth Types	This study aims to compare various cephalometric measurements and show the relationships between beta, W and Yen angles and the sagittal dimension of the maxilla and mandible in individuals with different vertical facial growth types.		150 lateral cephalograms with different types of vertical facial growth with low-angle (LA), norm-angle (NA), high-angle (HA) and Class I malocclusion. The following were assessed and compared with each other: ANB angle, Wits appraisal, A-Nperp, Pog-Nperp, Beta angle W angle Yen angle		The Kolmogorov–Smirnov and Shapiro-Wilk tests Levene’s test, analysis of variance, Kruskal–Wallis test, Mann–Whitney U test Spearman correlation test Statistical significance value was set as *p* < 0.05.	Analysis parameters of Wits appraisal, Pog-Nperp, Beta, W and Yen angles were significantly different among groups (*p* < 0.05). The Wits analysis, Pog-Nperp and Yen angles were found to be significantly lower in HA participants compared with LA participants, while the beta angle was found to be significantly higher in HA participants compared with LA participants (*p* < 0.05). Beta and W angles were significantly lower in NA patients than in HA patients (*p p* < 0.05). ANB, beta, W and Yen angles show significant correlations regardless of vertical face growth type (*p* < 0.05)	moderate
**(b)**									
**Q2**	**Authors (Year)**	**Title**	**Aim of the Study**	**Observers**	**Number of Participants**	**Studium Project**	**Statistical Method**	**Results According to Authors**	**Level of Evidence**
	2013	Rizwan M., Mascarenhas R., [16]	A new parameter for assessing vertical skeletal discrepancies: the R angle	The study aims to evaluate the reliability of R angle (nasion–centre of the condyle–menton) in assessing the vertical skeletal discrepancies.	80 patients aged 18–26 years	Evaluation of R angle in low-angle, average-angle and high-angle patient groups. Next, the R angle was individually constructed, measured and compared for each of the three skeletal patterns (high, average and low angle).	The means and standard deviations of R angle for all the three skeletal patterns were obtained using one-way ANOVA. The R angle values as examined by the Newman–Keuls post hoc test revealed that the three skeletal patterns under analysis are different.	Results: R angle < 70.50 indicates low- angle cases, between 70.5–75.50 indicates average-angle cases and > 75.50 indicates high-angle cases. R angle is clinically and statistically significant in assessing vertical skeletal discrepancies. Receiver operating characteristic (ROC) curves indicated that R angle > 70.50 had 81.6% sensitivity and 70% specificity in discriminating the low-angle cases from average-angle cases and R angle > 75.50 had 90% sensitivity and 77.8% specificity in discriminating the average-angle cases from high-angle cases. Therefore, values < 70.50 indicate low-angle cases, between 70.5–75.50 indicate average-angle cases and > 75.50 indicate high-angle cases.	moderate
	2016	Ahmed M, Shaikh A, Fida M. [25]	Diagnostic performance of various cephalometric parameters for the assessment of vertical growth pattern.	The Y-axis, sella–nasion angle to the mandibular plane (SN.MP), maxillary plane angle to the mandibular plane (MMA), sella–nasion to gonion–gnathion angle (SN.GoGn), Frankfort–mandibular plane angle (FMA), lower anterior facial height and total anterior facial height ratio (LAFH.TAFH) were used for assessing the vertical growth of the craniofacial region.	161 lateral cephalograms (71 men and 90 women) aged 23.6 ± 4.6 years The participants were divided into 3 groups: hyperdivergent, normodivergent and hypodivergent.	Comparisons: The sella–nasion angle to the mandibular plane (SN.MP), maxillary plane angle to the mandibular plane (MMA), sella–nasion to gonion–gnathion angle (SN.GoGn), Frankfort–mandibular plane angle (FMA), lower anterior facial height and total anterior facial height ratio (LAFH.TAFH).	Kappa statistics were used for comparing the diagnostic accuracy of different analyses. To further validate the results, sensitivity and positive predictive values (PPV) were calculated for each parameter.	SN.GoGn revealed significant intraclass agreement (k = 0.850). In the hypodivergent group, the highest sensitivity was shown by MMA (0.934) and the highest PPV (0.964) by FMA. In the normodivergent group, FMA showed the highest sensitivity (0.909) and the highest PPV (0.903) by SN.GoGn. SN.GoGn showed the highest sensitivity (0.980) and PPV (0.87) in the hyperdivergent group. SNGoGtn and FMA proved to be the most reliable indicators. LAFH and TAFH are the least reliable indicators for assessing the vertical growth pattern.	Moderate
	2018	Kattan EE., Kattan EM, Elhiny OA. [17]	A new horizontal plane of the head.	This study attempts to introduce a new extracranial horizontal plane of the head (plane K that extends from SN to SAE bilaterally) that could act as a substitute for the Frankfurt horizontal intracranial reference plane both clinically and radiographically.	A prospective study of 40 participants including 20 men and 20 women	The establishment of a stable anthropometric plane K compared with the Frankfurt plane when stabilised with the extraoral orientor for the determination of NHP	Descriptive statistics were used: mean, standard deviation and Student’s *t*-test.	The new plane K was found to be both reliable and reproducible. It can be used as a reliable reference plane instead of the Frankfort horizontal plane both clinically and radiographically; it is an accurate tool for head orientation in the natural head position.	Low
	2019	Park J.A., Lee J.S., Koh K.S., Song W.C. [18]	The use of the zygomatic arch as a baseline for clinical applications and anthropological research.	This study aims to establish a new cephalometric measurement to assess the skeletal jaw discrepancy using a new line and plane based on the landmarks of the zygomatic arch where each of them is the upper border. This line is in opposite to the Frankfurt plane.	170 adults aged 21–30 (100 men and 70 women)	The establishment of a more stable and easier to repeat finding of a landmark and horizontal plane compared with the Frankfurt plane.	The intraobserver and interobserver reproducibility of the angular measurement as well as side-related and sex-related differences were analysed using Student’s *t* test.	The horizontal plane through the Zy point was more stable than the Frankfurt plane. The angle between the Frankfurt plane and the plane through the upper border of the zygomatic arch was also constant: 4.5 degrees ± 2.5 degrees and ranging from −3.3 to 11.9 degrees.	low

## Data Availability

The data sets that are used and/or analysed during the current study are available from the corresponding author upon reasonable request.

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
