# Peer review of "New Sagittal and Vertical Cephalometric Analysis Methods: A Systematic Review"

_diagnostics, 2022, doi:10.3390/diagnostics12071723_

Round 1

Reviewer 1 Report

The work from Smolka et colleagues aims to systematically revise literature in order to deepen knowledge about "landmarks and reference linear and angular measurements of 2D cephalometric analyses 79 assessing the sagittal and vertical discrepancy in the position of jaw bases".

The manuscript is well written and well organized, and it meets all the scientific criteria for a systematic review protocol. It has a good scientific sound and it adds novelty to current literature knowledge. The paper well fits with the journal, so it deserves publishing in the present form.  

Author Response

Dear Reviewer. Thank you very much for your assessment and taking the time to review our work

Reviewer 2 Report

1. Please avoid using reference in the abstract.

2. The results showed that despite such a high number of articles published in peer-reviewed scientific journals, only 12 studies on new cephalometric analyses in the 26 sagittal plane and 4 studies on new cephalometric analyses in the horizontal plane met the criteria and, as a result, were included in the review........ there are multiple studies on sagittal and horizontal analysis are missing. Please continue searching and avoid such selection bias.

3. Try to conduct meta analysis and forest plot.

4. Please include figures.

Author Response

Dear Reviewer. Thank you very much for your time, in-depth analysis and constructive suggestions.
1.The references have been removed from the abstract.

2. Thank you very much for this attention. Nevertheless, we cannot agree with it. Choosing the right database search strategy is, in our opinion, a key element of any systematic review or meta-analysis. Indeed, resignation from the Risk of Bias analysis will undoubtedly increase the number of analyzed articles, but this is not the purpose of this type of article.
The aim of any systematic review, which can be concluded with a meta-analysis under favorable circumstances, is to critically evaluate the available data. To achieve this, in the first stage, the selection of articles should be carried out during the database search. Please note that in our case we refer to the systematic review published by Durao et all in Validity of 2D lateral cephalometry in orthodontics: a systematic revie.  Prog. Ortoda. 2013;14(1):31, extending it with new analyzes that have arisen since then, but maintaining the same search strategy.
This has been discussed by us in the article and in our opinion, firstly, guarantees a valid selection strategy that has already been reviewed and recognized, and also forms the basis for correct comparisons between subsequent reviews. When it comes to Risk of Bias analysis, we follow the Cochrane guidelines, recognizing it as the best and most widely recognized determinant of good practice in the field of data mining, systematic reviews and meta-analyzes.
Nevertheless, we are aware that the greatest sin of a researcher is to over-believe in his own infallibility. So we will sincerely appreciate you for submitting the details of the articles we missed. If they meet the requirements of a sufficiently thorough and high-quality selection, we will certainly take them into account in our analysis and we will be grateful for your help in improving the quality of our work. 3. Thank you very much for this attention. To our knowledge, performing a meta-analysis does not only consist in performing statistical tests and pasting a graph. Meta-analysis, if it is to be a scientific work of an appropriate quality, requires meeting a number of requirements. The sense of performing a systematic review is precisely to analyze whether the available studies are of appropriate quality and meet the requirements for studies qualified for the meta-analysis. If our systematic review allowed us to collect such a group of studies, we would definitely perform a meta-analysis, because it is the crowning achievement of the data mining process. Unfortunately, our review did not allow this, and this is the most important conclusion of this review - a hint for researchers that research on this topic should be continued because more original studies are needed with a low risk of methodological error. 4.Thank you for your attention What are the figures?

Reviewer 3 Report

This subject is not correctly presented : nowadays, an orthodontic diagnosis is often made on a CBCT imaging of the patient's head... The here proposed comparison should be more effective if the authors could compare the 2D measurements done on lateral cephs to measurements done on CBCT. This is a systematic review on new planes and points plotted on lateral ceps, but nothing is said about radioprotection (many papers reported on quantification of radiation in lateral cephs and CBCT..) nor about the eventual apport of frontal cephs ton an orthodontic diagnosis....This at least should be mentioned in the discussion...

In the legends of tables 2 and 3, "red coloured.." may be deleted, because there are no red coloured circles in these tables.....

Two of the reported studies seem to this reviewer without any  interest in this paper : the references to Kattan et al. (2018) and to Park et al. (2019) don't add anything to the paper. These authors mention new planes but no new measurements. If You take these two papers out of the study, only 2 papers with new horizontal measurements remain, and then it is of course not possible to draw any conclusions of only two papers in a systematic review...

This reviewer suggests also to add some illustrations on the new points and angles, in order to facilitate their comprehension...

Author Response

Dear Reviewer. Thank you for your attention. 

  1. In the discussion section we are added the appropriate sentences.

Nowadays, the increasing use and availability of CBCT equipment is largely related to the issue of radiological protection. CBCT is associated with higher radiation dose than OPG or cephalogra. Howevew, if one projection provides the possibility of solving several diagnostic problems, it will allow for registration from making several projections in favor of one CBCT image. However, in order to be able to perform cephalometric analyzes on the CBCT projection, it is necessary to evaluate the accuracy  of introducing points is sucha n image and above all, to develop cephalometric analyzes intended for such imaging. As long as such analyzes and studies are created it will be possible to refer tchem, among others, to the presented systematic review in order to compare the diagnostic value of 2D and 3 D cephalometric analyzes.

  1. I n tables 2 and 3 the red colour has been removed
  2. Thank you for your attention. The presented reaserch takes into account the introduction of new planes wich in itself affects the messurements made. Since each angular or specific planes, it follows from the principles of geometry that the metod of their determination affects the vallue and accuracy of such a messurement. For this reason, the autors decided to include these studies in the analysis. Moreover , they met the requipments of the risk of bias analisis. According to our analysis, these are well-designed studies that discuss the issue of determining the plane for horizontal measurements. For this reason, they were included in the analysis.

In terms of the numer of articles available, the small numer of well-designed studies could be one of the systematic review, To our knowleadge, metaanalysis in the crowning achievement of a research proces on a given topic, conducted by various researchers. However, in order to be able to perform a meta-analysis, it is neccesary to collect a sufficiently large numer of original studies with a low risk of research error. To answer the question of whether such studies are available  at all, a systematic review must first be performer, which may show that such studies are available but have a risk of research error, ort hey simply do not exist. The conclusion can therefore be that such studies must be performer and/or that they have to be performer in view of the suggestions regarding  the risk of research error. In our review we are dealing with both conclusions.

So we ask the Reviewer to take into account our explanation.

Round 2

Reviewer 3 Report

Thnaks for the comments and changes, but "Red colours.." remain in the legend of the Table 3b and must be removed !

Author Response

Dear Reviewer. Thank you very much for your attention. The red color has been removed from Table 3b.
